

# Assessing migration patterns in *Passerina ciris* using the world's bird collections as an aggregated resource

Ethan Linck[1], Eli S. Bridge[2], Jonah M. Duckles[3], Adolfo G. Navarro-Sigüenza[4] and Sievert Rohwer[1]

[1] Department of Biology and Burke Museum of Natural History & Culture, University of Washington, Seattle, WA, United States
[2] Oklahoma Biological Survey, University of Oklahoma, Norman, OK, United States
[3] The Software Carpentry Foundation, Norman, OK, United States
[4] Departamento de Biología Evolutiva, Museo de Zoología, Universidad Nacional Autónoma de México, México DF, Mexico

## ABSTRACT

Natural history museum collections (NHCs) represent a rich and largely untapped source of data on demography and population movements. NHC specimen records can be corrected to a crude measure of collecting effort and reflect relative population densities with a method known as abundance indices. We plotted abundance index values from georeferenced NHC data in a 12-month series for the new world migratory passerine *Passerina ciris* across its molting and wintering range in Mexico and Central America. We illustrated a statistically significant change in abundance index values across regions and months that suggests a quasi-circular movement around its non-breeding range, and used enhanced vegetation index (EVI) analysis of remote sensing plots to demonstrate non-random association of specimen record abundance with areas of high primary productivity. We demonstrated how abundance indices from NHC specimen records can be applied to infer previously unknown migratory behavior, and be integrated with remote sensing data to provide a deeper understanding of demography and behavioral ecology across time and space.

## INTRODUCTION

Natural history museum collections (NHCs) represent a rich and largely untapped source of data on demography, behavioral ecology, and population movements (*Ricklefs, 1997*; *Krosby & Rohwer, 2010*; *Suarez & Tsutsui, 2004*). Housed in museums and herbaria worldwide, NHCs are unique among extant biological datasets in their breadth and depth, and they lack some of the biases intrinsic to data collected for a specific research goal. NHCs are particularly valuable in that the oldest specimens in collections predate even the longest running ecological surveys (*Magurran et al., 2010*), and the majority of specimens are associated with detailed provenance data (*Lister & Group, 2011*). The combination of these

Corresponding author
Ethan Linck, elinck@uw.edu

records into sortable databases spanning multiple institutions provides an invaluable resource in approaching a wide range of biological questions.

NHCs have traditionally been used to assess biogeographic range changes (*Boakes et al., 2010*), phenological shifts (*Robbirt et al., 2011*), hybridization (*Rohwer & Wood, 1998*) and evolutionary change in morphology (*Hromada et al., 2003*). Applications of molecular techniques to NHCs have extracted DNA from historic specimens to use in phylogenetic analyses (*Paabo et al., 2004*), performed stable isotope analyses to track diet and migration in birds (*Inger & Bearhop, 2008*), and examined environmental contamination through trace element analysis (*Berg et al., 1966*; *Hickey & Anderson, 1968*). Specimen collections similarly have the potential to shed light on population dynamics, but only if information on collecting effort intrinsic to these data is available.

One method of overcoming this shortcoming is the application of abundance indices for a particular species in which the number of NHC specimens representing the species is corrected to a crude measure of effort (*Miki et al., 2000*; *Barry et al., 2009*). This crucial effort measure can be generated from electronic natural history museum catalogs (such as VertNet.org) by aggregating records of specimens from a particular region and time period that are expected to have been collected in a similar manner to the focal species of a study. Abundance indices have been successfully applied to show molt migration (*Barry et al., 2009*), population dynamics in medicinal plants (*Miki et al., 2000*), migratory double-breeding (*Rohwer et al., 2012*), and changes in community composition from massive environmental pertubations (*Rohwer, Grason & Navarro-Siguenza, 2015*). A logical extension of these analyses is to examine spatial and temporal changes in abundance index values to infer month-to-month population-level movements, which has historically been difficult for small, highly mobile species such as migratory birds. For these species, technology, cost and unpredictable behavior often prohibit direct tracking of individuals. However, using aggregated collection records to quantify spatio-temporal variation in population dynamics remains untested.

Here, we demonstrate how abundance indices can be applied to infer population-level movements from across the non-breeding range of a migratory passerine, the Painted Bunting (*passerine ciris*). We plotted abundance index values from georeferenced NHC data in a 12-month series for the Midwestern US breeding population of this new world migratory passerine across its molting and wintering range in Mexico and Central America. We found a statistically significant change in abundance index values across regions and months that suggests a quasi-circular movement around its non-breeding range, and we linked this movement pattern to the phenology of plant growth in Mexico as determined by the enhanced vegetation index (EVI) of primary productivity (see *Matsushita et al., 2007*).

## METHODS

### Focal taxon

The Painted Bunting (*Passerina ciris*) is a migratory New World passerine in the family Cardinalidae. Current taxonomy recognizes two subspecies of Painted Bunting but the boundary between these races does not coincide with a nearly 500 km gap separating the

east coast and the Midwestern breeding populations of Painted Buntings (*Thompson, 1991*). Further, these isolated breeding populations differ dramatically in their molt scheduling, with the eastern population molting on its breeding range prior to migration and the Midwestern population moving to the monsoon region of the southwestern United States and northwestern Mexico where it pauses to molt before proceeding to its wintering range in southern Mexico and Central America (*Thompson, 1991*; *Rohwer, Butler & Froehlich, 2005*; *Rohwer, Rohwer & Ramírez, 2009*). In this study, we focus exclusively on the Midwestern breeding population of Painted Bunting.

Across their range, Painted Buntings favor ecotones with brushy, weedy habitats in second growth, and dense forest understory. Relatively little is known about the species' movements following molt stopover, but progressive southward movements of populations along the west coast of Mexico have been observed in the autumn (S Rohwer, pers. comm., 2014; *Contina et al., 2013*).

## Calculating abundance indices

To track spatial and temporal changes in Painted Bunting population densities during the wintering season, we employed a method of inferring relative population densities from specimen collections data known as Abundance Indices. The method, proposed in *Rohwer et al. (2012)* and developed independently by *Miki et al. (2000)*, adjusts for a major shortcoming of specimen collections data—the absence of associated information on collecting effort—by producing an index that is corrected to a crude measure of effort. To produce an abundance index, electronic natural history museum catalogs (such as VertNet.org) are used to aggregate NHC specimen records of (1) the species of interest and (2) other taxa that are typically collected with similar methods to the species of interest. Raw counts are then used to determine the proportion of focal species specimens to all collected specimens from a particular region and time period, allowing for comparisons of abundance across regions with different histories of collecting effort.

We used the formula for abundance index calculation proposed in *Rohwer, Grason & Navarro-Siguenza (2015)*:

$$AI_{kr} = 100 \frac{\chi_{kr}}{\sum_{j=1}^{n} \chi_{jr}}$$

where $\chi_{kr}$ is number of specimens of the kth species collected in $r$, the region and time period of interest, and $n$ is the number of specimens of all species that would be "expected" to be collected using the same methods in that region and time period of interest.

## Reference data

In order to calculate abundance indices for Painted Buntings, we accessed two databases of specimen collection records: the Mexican Bird Atlas, and VertNet. The Mexican Bird Atlas began compilation by A. Navarro and T. Peterson in the 1990s, and now represents the most complete reference of study skins of Mexican birds residing in natural history museums worldwide (*Navarro-Siguenza, Peterson & Gordillo-Martínez, 2003*). The Atlas now contains records of more than 370,000 specimens from 71 museums, and is completely

georeferenced (meaning each specimen record is associated with the latitude and longitude of its collection locality). We used records for the Mexican Bird atlas for all indices calculated within the political boundaries of Mexico. The VertNet data portal (VertNet.org) is an NSF-funded collaborative project to make biodiversity information, including specimen collections records, freely and easily accessible to the public. We used records from VertNet to examine raw bunting counts by month for the Central American countries of Guatemala, El Salvador, Honduras, Nicaragua, Costa Rica, and Panama.

## Subsetting and data cleaning

We produced abundance indices for Painted Buntings for each month of the year. These indices were produced on a relatively fine spatial scale for Mexico (a 5 min latitude by 5 min longitude grid), but were only produced on a country-wide level for Central America, due to the limited number of properly georeferenced records in the VertNet data. To calculate abundance indices, we referenced Painted Bunting collections against the combined records of species collected using similar methods. We follow *Rohwer et al. (2012)* in including other taxa commonly collected with mist-nets and small-bore shotguns: Passerines (order Passeriformes), Cuckoos (order Cuculiformes), and Woodpeckers (order Piciformes; family Picidae). While abundance index values are sensitive to reference specimen selection and collection bias (e.g., collectors disproportionately targeting rare species), we believe the high species diversity and large number (>245,000) of specimens included in our reference specimen database minimize the influence of this shortcoming on our conclusions.

Despite the fact that majority of specimen collections records accessed from the Mexican Bird Atlas were both dated and georeferenced, a subset (<10% in both Painted Bunting and reference specimen data) had either missing or obviously erroneous values for date or latitude and longitude coordinates. These were excluded from all subsequent analyses.

## Analyzing migration patterns

We used a Geographic Information Systems (GIS) approach to plot all specimen collections records from the Mexican Bird Atlas, for both Painted Buntings and reference specimens. A 5-minute raster grid was initially overlayed on plotted reference specimens, which were then transformed into a scaled map of all collected specimens in a particular region and month. In any grid square where Painted Bunting specimens were collected, an abundance index was calculated and plotted as a circle, its diameter proportional to the value of the index. Although abundance indices were produced for Central America, we did not incorporate these into our geospatial analysis due to exceedingly few Painted Bunting specimens and corresponding low AI values.

To determine the statistical significance of any observed patterns of spatial and temporal change, we divided Mexico into three regions corresponding with contiguous bands of Painted Bunting habitat, (NW, NE, and S, defined by the 20th parallel north and the 103rd meridian West, respectively; Fig. 1). Among these regions, we performed three Pearson's chi-square tests for changes in abundance indices in Painted Buntings and reference specimens during three time periods: the molt-stopover period (July–October), winter (November–February), and spring migration (March and April). Specifically,

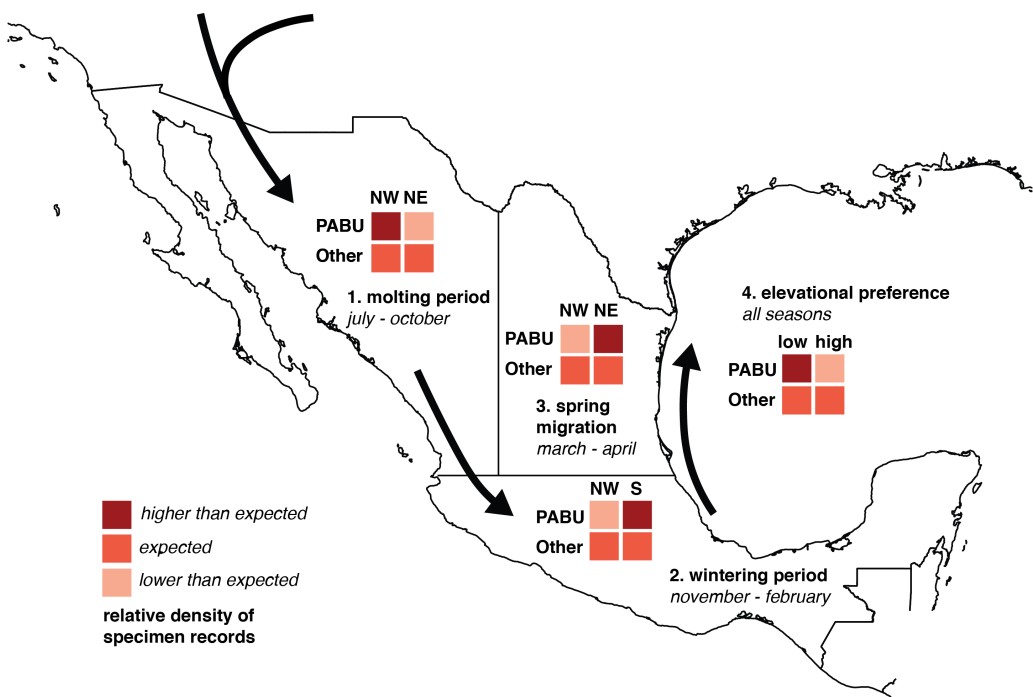

**Figure 1** **Diagram of chi-square test predictions of seasonal changes in Painted Bunting specimen record densities.** Chi-square analysis of Painted Bunting population movements around Mexico. The 2 × 2 grids illustrate predictions of relative values in Pearson's chi-square contingency tables, testing whether Painted Bunting ("PABU") populations were significantly greater in a particular region (NW, NE, and S, divided by 20° N and 103° west and marked on the plot) and a particular time (molting period, wintering period, and spring migration) than expected with respect to reference specimen populations ("Other").

we asked (1) whether Painted Bunting records were significantly more numerous than expected by chance (relative to reference specimens) in NW Mexico than NE Mexico during the molting season; (2) whether Painted Bunting records were significantly more numerous than expected by chance in Southern Mexico than NW Mexico during the winter; and (3) whether Painted Bunting records were significantly more numerous than expected by chance in NE Mexico (along the Gulf of Mexico) than in NW Mexico during spring migration. To account for the possibility that migrants from the Southeastern US population of Painted Buntings were wintering on the Yucatán Peninsula and thus inflating specimen densities to the extent that they influenced statistical significance of our results, we repeated our comparison of records in Southern Mexico and NW Mexico excluding all records east of the Isthmus of Tehuantepec (at 94° West). Additionally, to evaluate a hypothesis based on observations by S Rohwer (pers. comm., 2015) that Painted Buntings were primarily restricted to coastal lowlands during the non-breeding season, we performed a chi-square test to determine whether Painted Bunting records were significantly more numerous below 200 m in elevation. We display contingency tables for these tests and our predictions for relative specimen densities consistent with a pattern of circular movement around coastal Mexico in Fig. 1. This aggregated measure of abundance change allowed us to rigorously test our interpretation of the direction of migration.

Finally, to provide additional ecological context to our findings, we investigated the correlation between bunting abundance and primary productivity. We downloaded monthly means for the Enhanced Vegetation Index (EVI) compiled from 2000 to 2010 from the North American Vegetation Index and Phenology Lab website (http://vip.arizona.edu). We used EVI, as opposed to the more widely used Normalized Difference Vegetation Index (NDVI), as an index of primary productivity because of EVI's enhanced sensitivity in high biomass regions (such as the Painted Buntings' wintering sites) and its robustness against atmospheric influences (*Liu & Huete, 1995*; *Matsushita et al., 2007*). The downloaded data for each monthly mean consisted of a georeferenced HDF raster file at a 0.05° resolution. We extracted the EVI data layer and clipped it to the area of interest (Latitude: 10 to 40°, Longitude: −125 to −70°).

For each month, we extracted EVI values for pixels within a 10 km radius of each collection site. We included data from each specimen such that locations from which multiple specimens were collected were represented multiple times in the data set. We assume that collection sites that yielded multiple birds are indicative of the most suitable or desirable habitat for Painted Buntings, and that they should be weighted in a corresponding manner when evaluating the relationship between EVI and Painted Bunting distributions.

To test the simple null hypothesis that specimen locations for Painted Buntings were random with respect to EVI, we generated 500 uniformly random locations within the borders of Mexico and repeated the extraction process described above with each monthly EVI map and the 500 random points. We averaged the pixels from each location and then compared the set of EVI values for each month from the specimen locations to the corresponding EVI values associated with the random locations. We performed a *t*-test for each monthly data set and calculated 95% confidence intervals for each overall mean.

Because randomly chosen points are not necessarily a good representation of available habitat for Painted Buntings, and because of non-independence of some Painted Bunting collecting sites (those that are repeated in the analysis) we generated what is arguably a more comparable set of reference locations by randomly choosing 250 locations for each month from the museum reference data described above. This subsampling process began by rounding the coordinates (latitude and longitude) of all locations to the nearest 0.01° (about 1 km). We then sampled 250 unique locations from the specimen data, and we dervied EVI values for each location as described above. We filtered out locations where there were fewer than three resulting EVI pixels within the buffer area (such low pixel counts can result from sites surrounded by null pixels associated with water bodies). We then duplicated each EVI score according to how many times it was represented in the entire data set for the relevant month. These subsamples allowed us to effectively weight each location based on the total number of birds collected there in a manner similar to the weighting of the Painted Bunting collection that resulted from repeated location data.

Initial manipulation of EVI data was performed using the gdal translator library (http://www.gdal.org). All subsequent analysis were performed in R version 3.1.0 (*R Core Team, 2014*) using the following packages: raster (*Bivand & Rundel, 2015*), maptools (*Bivand & Lewin-Koh, 2015*), plyr (*Wickham, 2011*), rgeos (*Bivand & Rundel, 2015*), and ggplot2 (*Wickham, 2009*).

**Table 1** **Painted Bunting abundance index values by month and region (NW, NE, and S, divided by 20° N and 103° west).** Values over 0.5 are bolded, highlighting periods of relatively high abundance for a given area.

|  | NW | NE | S |
| --- | --- | --- | --- |
| January | 0.4027 | **0.9969** | **0.9405** |
| February | 0.3362 | 0.3296 | **0.798** |
| March | 0.2055 | 0.4709 | **1.146** |
| April | 0.1513 | 0.4198 | **0.619** |
| May | 0.0333 | **0.7953** | 0.1048 |
| June | 0.0269 | 0.1707 | 0.0328 |
| July | **1.4121** | 0.1699 | 0 |
| August | **2.9319** | 0.2691 | 0.0193 |
| September | **3.8916** | 0.1466 | 0.0771 |
| October | 0.3048 | 0.2214 | **0.6979** |
| November | 0.1358 | 0.829 | **0.6141** |
| December | 0.2685 | **0.5596** | **0.9829** |

## RESULTS

### Migration analysis

Our plotted monthly abundance indices for *P. ciris* confirm a pattern of population-level movement across Mexico throughout the year (Fig. 2 and Table 1). AI values plotted for July illustrate an east–west split during mid-summer, with high AI values forming two clusters in Northern Mexico: an eastern cluster in Nuevo Leon and Tamalpais, and a western cluster in Sinaloa and Durango. In August and September, these associations persist, with the western cluster increasing both by number of raster grid squares reporting an abundance index, and by value of plotted abundance indices. October, November, and December show the southward movement and diffusion of plotted AI values on both coasts of Mexico. Abundance indices again hug the states of both coasts, forming a loose western cluster in Guerrero, Michoacán, Oaxaca, Jalisco, and Colima, and a loose eastern cluster in the Veracruz, Tabasco, Campeche, and Yucatån. Although there is then no observable pattern in plotted AI values from January to February within or among these clusters, this period is followed by a strong association of AI values in northeast Mexico (Coahuila, Nuevo Leon, Tamalpais) and an absence of values elsewhere in the months of March and April. South of Mexico, specimen records indicate the presence of Painted Buntings at extremely low densities, mostly restricted to the winter months of November to March. Pooled raw counts of buntings for all records in this region (including Belize, El Salvador, Honduras, Guatemala, Nicaragua, Costa Rica, and Panama) confirm the near absence ($n < 10$ per month) of bunting specimens collected during the July–October stopover period (Fig. 3).

### Statistical tests

Our chi-sq tests confirm significantly higher Painted Bunting record abundance than expected for all four analyses. Painted Buntings were (1) significantly more numerous
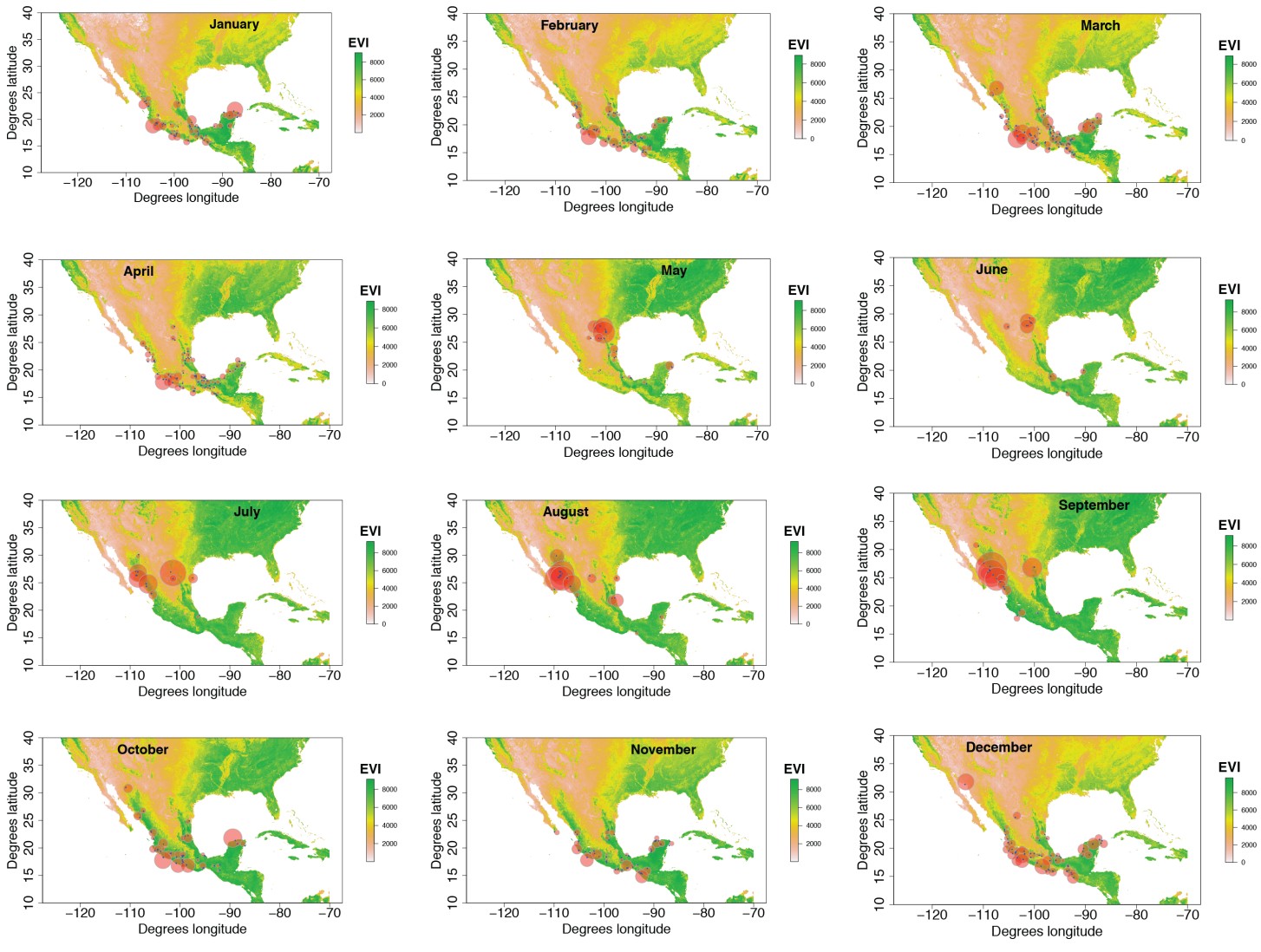

**Figure 2** **Monthly changes in Painted Bunting abundance index values with EVI analysis of remote sensing data.** (also provided as .gif animation in Supplemental Information 1). Abundance index (AI) values for Painted Bunting specimens in Mexico by month, plotted against EVI analysis of remote sensing data. Red circles indicate the occurrence of Painted Bunting specimens, with the diameter of the circle proportional to AI value. Green areas indicate high EVI values, correlated with regions with a high density of live green plants (photosynthetically active vegetation).

in NW Mexico than NE Mexico during the molt-stopover period compared to reference specimens (question 1; $X$-squared $= 108.8812$, $df = 1$, $p$-value $< 0.0005$), (2) significantly more numerous in southern than NW Mexico in the winter than reference specimens (question 2; $X$-squared $= 46.6711$, $df = 1$, $p$-value $< 0.0005$ including Yucatán specimens; $X$-squared $= 48.4392$, $df = 1$, $p$-value $< 0.0005$ excluding Yucatán specimens), (3) significantly more numerous along the Gulf of Mexico than along the west coast of Mexico, compared to reference specimens during spring migration (question 3; $X$-squared $= 12.4593$, $df = 1$, $p$-value $< 0.0005$), and (4) significantly more numerous below 200 m elevation (question 4; $X$-squared $= 399.5081$, $df = 1$, $p$-value $< 0.0005$).

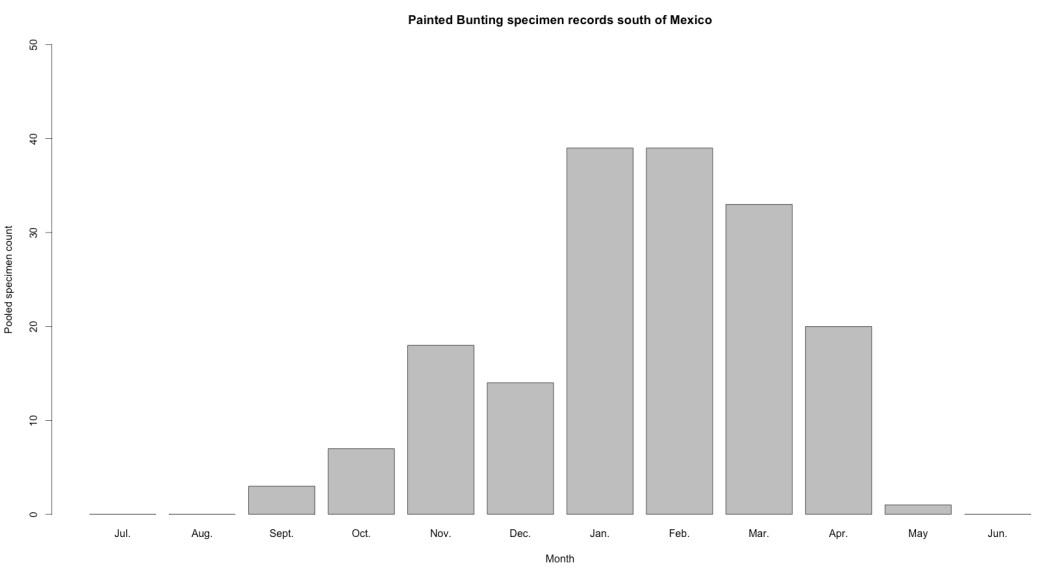

**Figure 3  Histogram of raw Painted Bunting specimen counts in Central America.** Raw Painted Bunting specimen records pooled from Belize, Guatemala, El Salvador, Honduras, Nicaragua, Panama, and Costa Rica, and totaled by month of collection.

## Remote sensing

The EVI data extractions for Painted Bunting collection sites yielded an average of 10.4 pixels per location (range = 3–14). Locations near coastlines or on islands often had fewer pixels than inland site as the EVI data did not extend into water bodies. Extractions from random locations yielded an average of 11.1 EVI pixels (range: 4–16) and for the subset of reference specimen location data we got an average of 10.3 pixels (range: 3–14). After filtering, the reference EVI data consisted of a minimum of 1,939 values for each month from a minimum of 245 unique locations. For the real data, the number of unique locations ranged from 9 (in June) to 98, and the number of EVI values ranged from 11 (also in June) to 219.

For 10 months of the year, Painted Bunting collection sites in Mexico had higher EVI scores (i.e., higher primary productivity) than randomly generated locations within Mexico ($p < 0.01$; Fig. 4A). The only exceptions were May and June, when Painted Buntings are on their breeding grounds and are relatively scarce in Mexico. The highest monthly EVI average associated with the specimen data was from the month of October, which corresponds with high Painted Bunting densities in the states of Sinaloa and Sonora, where many if not most Painted Buntings undergo their annual molt (*Rohwer, 2013*). It is also in the month of October that we observed the greatest difference between the mean EVI value for collection sites and for random sites.

Comparison of EVI scores for Painted Bunting collection sites with those of other collecting sites indicated selection by the buntings for high productivity areas in only 5 months of the year (October, November, February, March, and April; Fig. 4B). In the months of July and September, EVI scores were significantly higher ($p < 0.01$) for the reference data. The discrepancies in the results from this analysis and the one based on

                                        

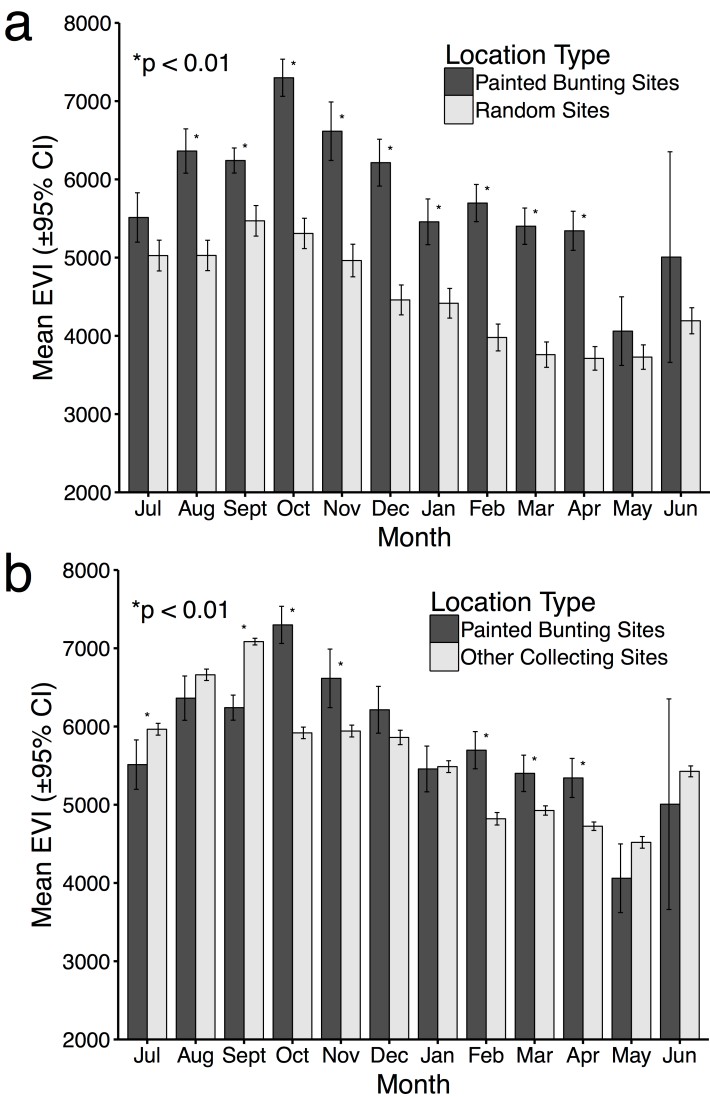

**Figure 4  Mean EVI values of specimen data compared to mean EVI values of randomly distributed points, by month.** Mean EVI values of Painted Bunting specimen records compared to (A) mean EVI values of 500 randomly distributed points within Mexico; and (B) mean EVI values of 250 points sampled from reference specimen collection localities.

random locations is likely due to a general collection bias toward highly-productive sites which are likely to have increased bird abundance and diversity.

## DISCUSSION

The spatial and temporal changes in plotted abundance indices presented in Fig. 2 illustrate that an abundance index approach can be applied to NHC datasets to infer population-level movements across a species' range from month to month. The advantages of this approach in determining general trends within or among taxa are numerous. Analyzing spatial and temporal changes in abundance indices allows for the repurposing of a comprehensive and pre-existing source of species occurrence data into a tool for investigating questions about

behavior and population movement. In doing so, the approach offers a complement to geologger tagging studies (*Contina et al., 2013*), which can often be costly and logistically difficult. Perhaps most importantly, the use of NHC datasets allows for the potential of describing *historical* population-level movements, phenomena that might otherwise go undescribed due to an absence of contemporary observers, and the disturbance of decades of anthropogenic pressure on populations and land-use change that may have changed historic movement patterns.

Our results also shed light on previously unconfirmed migratory behavior in *P. ciris.* The initial clustering of high AI values in July–September in northwestern Sinaloa (Fig. 2) corroborates evidence of molt-migration stopover in agricultural habitats in NW Sinaloa for subspecies *P. c. pallidor* (*Contina et al., 2013*; *Rohwer, 2013*; *Rohwer, Rohwer & Ramírez, 2009*; *Rohwer, Butler & Froehlich, 2005*). We believe subsequent southward progression and diffusion visible in abundance index values across the southern half of Mexico from October to February is consistent with anecdotal observations by S Rohwer (pers. comm., 2015) describing a complete absence of wintering Painted Buntings in regions where they had been previously been abundant during the molting period, as well as geologger tag and isotope evidence from *Contina et al. (2013)* of similar movement. The limited number of specimen records elsewhere in Central America provides additional support for this movement, as Painted Buntings are largely unrecorded south of Mexico until well after the their molt stopover period in NW Mexico (Fig. 3). A reduction in individual grid-square AI values and increase in overall number of grid squares filled also correlates with expected migratory behavior. Finally, plotted AI values in March and April illustrate the high population densities in NE Mexico in the Gulf Coast migration corridor to be expected during spring migration through this region to breeding grounds in the United States.

Taken in sum, monthly plotted abundance indices (Fig. 2) indicate a quasi-circular movement of *P. ciris* populations around coastal and Southern Mexico. We believe these patterns can be partially explained by the EVI analysis of remote sensing data presented in Fig. 3. A period of peak live green vegetation in Mexico in the months of July–September correlates with the cluster of abundance indices representing the molt-migration stopover site in Sinaloa for the Midwestern breeding population identified for the same period in Fig. 1. After a period of reduction in green vegetation from October–February, a second peak in live green vegetation in NE Mexico correlates with an increase in population densities of Painted Buntings along the NE coast of Mexico immediately prior to spring arrival on their principle midwestern breeding grounds in the United States.

EVI plots indicating peaks in live green vegetation can be thought of as a rough indicator of primary productivity and corresponding resource availability. *P. ciris* population densities therefore appear to shift in tangent with precipitation and plant growth, a logical correlation given *P. ciris* feeds primarily on grass seeds during the winter, and supported by our comparison with randomly generated localities. The comparison of EVI data associated with collection sites and randomly generated sites (Fig. 4A) confirms that the dynamic distribution of Painted Buntings as evinced by museum collection data corresponds in a non-random manner with increased primary productivity across the landscape. The discrepancies between the comparisons with random locations (Fig. 4A)

and subsets of the specimen reference data (Fig. 4B) revealed evidence of collection bias for areas of high productivity. This bias is not surprising given as we might expect collectors to work in productive areas that feature an abundance of birds, but it points to a shortcoming of most specimen-derived datasets—i.e., a lack of systematic or truly random sampling. Nevertheless, numerous studies have documented similar associations between migration routes and primary productivity, including studies of Painted Buntings (*Bridge et al., 2015*) and various tests of the green-wave hypothesis (*Drent, Ebbinge & Weijand, 1978*; *Owen, 1980*; *Shariatinajafabadi et al., 2014*; *Si et al., 2015*). Therefore, we present this finding as validation that our specimen based distribution mapping confirms expected patterns, rather than a novel correlative observation. Likely also due to resource limitation, our finding that Painted Bunting specimen records were significantly more numerous below 200 m supports claims that *P. ciris* primarily winters in the lowlands (S Rohwer, pers. comm., 2015; *Howell & Webb, 2007*).

Although we demonstrate the utility of NHC abundance indices in inferring population level movements, we reiterate that the technique in no way reflects the movements of individual birds. AI values represent stationary population densities at a particular time and place, and as such, caution must be taken not to over interpret findings, while keeping an open mind to alternate hypotheses. These include the existence of sedentary populations with geographically distinct distributions, and the potential of results being an historical artifact of a particular collecting expedition in regions with limited collecting effort. However, assuming thorough background collecting, the absence of target species at a particular time and place almost certainly represents the mass movement of individuals (rather than huge die-offs). In light of this, we believe the method can be applied to significantly more complex cases than the one described above. We believe the relative strengths and weaknesses of NHC abundance indices can complement similar studies (e.g., *La Sorte et al., 2016*) of population level movements implemented using citizen science data (such as eBird; http://ebird.org). While NHC data may be more sensitive to collecting bias and more limited in sample size than citizen science data (as discussed above), it offers advantages in increased temporal scope and better coverage in developing nations where citizen science initiatives and amateur natural history (as a widespread pastime) are still in their infancy.

We are particularly interested to see studies with well-sampled species in regions where anthropogenic disturbance has substantially altered migratory corridors in recent years. We hope in the future AI values will shed light on avian demographics, behavior, and distribution, and continue to illustrate the immense value of NHCs worldwide.

## CONCLUSIONS

Our study illustrates the utility of NHC specimen collection records in inferring population-level movement through abundance index analysis. We find evidence of quasi-circular movement from month to month in *Passerina ciris* populations across its non-breeding range, with abundance index values non-randomly distributed in regions with high EVI values (indicating high primary productivity).

## ACKNOWLEDGEMENTS

Thanks to John Klicka, Cooper French, Dave Slager, CJ Battey, and Kevin Epperly for reading and providing helpful feedback on the manuscript. Thanks to Alejandro Gordillo for coordinating geolocation of specimen records. Additionally, thanks is due to the following collections for specimen records in the portion of the Mexican Bird Atlas used for this study: American Museum of Natural History; Academy of Natural Sciences of Philadelphia; Bell Museum of Natural History, University of Minnesota; Natural History Museum (Tring, UK); Zoologische Forschungsinstitut und Museum Alexander Koenig; Übersee-Museum Bremen; Carnegie Museum of Natural History; California Academy of Sciences; Canadian Museum of Nature; Coleccion Ornitologica Centro de Investigaciones Biológicas UAEM; Cornell University Museum of Vertebrates; Denver Museum of Natural History; Delaware Museum of Natural History; Fort Hays State College; Field Museum of Natural History, Senckenberg Museum Frankfurt, Colección Nacional de Aves Instituto de Biología UNAM; Instituto de Historia Natural y Ecología; University of Kansas Natural History Museum; Los Angeles County Museum of Natural History; Natuurhistorische Musem Leiden; Louisiana State University Museum of Zoology; Museo de las Aves de México; Museum of Comparative Zoology; Moore Laboratory of Zoology; Museum Nationale D'histoire Naturelle Paris; Zoological Museum Moscow State University; Museum Mensch und Natur Munich; Museum of Vertebrate Zoology; Museo de Zoología Facultad de Ciencias UNAM; Royal Ontario Museum; San Diego Natural History Museum; Staatliche Museen fur Naturkunde, Stuttgar;, Southwestern College, Kansas; Universidad Autónoma de Baja California; University of Arizona; University of British Columbia Museum of Zoology; University of California Los Angeles; Florida Museum of Natural History; University of Michigan Museum of Zoology; United States National Museum of Natural History; University of Washington Burke Museum; Western Foundation of Vertebrate Zoology; and Yale University Peabody Museum.

### Funding

Support for fieldwork in northwestern Mexico came from the Burke Museum Endowment for Ornithology, and grants from Hugh S. Ferguson, the Nuttall Ornithological Club, and Region 6 of the USFWS. The funders had no role in study design, data collection and analysis, decision to publish, or preparation of the manuscript.

### Grant Disclosures

The following grant information was disclosed by the authors:
Burke Museum Endowment for Ornithology.
Nuttall Ornithological Club.
USFWS.

### Competing Interests

The authors declare there are no competing interests.

## Author Contributions

- Ethan Linck performed the experiments, analyzed the data, contributed reagents/materials/analysis tools, wrote the paper, prepared figures and/or tables.
- Eli S. Bridge conceived and designed the experiments, performed the experiments, analyzed the data, contributed reagents/materials/analysis tools, wrote the paper, prepared figures and/or tables.
- Jonah M. Duckles conceived and designed the experiments, performed the experiments, analyzed the data, contributed reagents/materials/analysis tools, prepared figures and/or tables.
- Adolfo G. Navarro-Sigüenza contributed reagents/materials/analysis tools.
- Sievert Rohwer conceived and designed the experiments, contributed reagents/materials/analysis tools, wrote the paper, reviewed drafts of the paper.

## Data Availability

Data and code are available via two Github repositories: PABU (for statistical tests and raw data, https://github.com/elinck/PABU) and mexpabu (for GIS and EVI analyses and raw data; https://github.com/Eli-S-Bridge/mexpabu).

## Supplemental Information

Supplemental information for this article can be found online at http://dx.doi.org/10.7717/peerj.1871#supplemental-information.

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
