# Peer review of "Assessing migration patterns in Passerina ciris using the world’s bird collections as an aggregated resource"

_PeerJ, doi:10.7717/peerj.1871_

## Round 0.1 · original submission · Major Revisions

Both reviewers find the study an interesting and a potentially valuable contribution. However, in a very careful and detailed review, Reviewer 1 raises a number of important issues regarding the proposed methodology. The reviewers raises concerns about whether the methods proposed by the authors are sensitive to biases that may be introduced by low abundance, as well as the sampling procedure used to quantify EVI. The authors will need to show that their results are robust. The reviewer also has a number of other minor points, which need to be addressed.

Also be sure to notice the attached PDFs from both reviewers.

Reviewer 1 ·

Basic reporting

Overall, the basic reporting of the article is good. The text is generally well-written, with good introduction and background. I have included some minor suggestions to improve the writing in the attched manuscript.

The figures could be improved. In particular, the contingency tables in figure 1 are confusing. The text does not mention testing NW vs NE (or NW vs S below) but the table suggests these areas were compared. Also, it seems strange that the boxes are empty. Should they contain some predictions? Overall, I think this figure needs to be rethought. I like the idea of showing the predicting molt-migration route but the tables are distracting and confusing. Consider replacing with something more intuitive and be sure all labels match the text.

As mentioned below, the specimen data and simulated data potentially have very different distributions, given the non-independence of the specimen data. Unfortunately, bar plots used in figure 3 hide these underlying distributions, making it difficult for readers to assess the structure of the data. I would recommend using a different type of plot to show these results. See Weissgerber et al. 2015 PlosBiology (DOI: 10.1371/journal.pbio.1002128) for examples of alternatives

Lastly, it would be helpful the display the AI summaries for each month/region in tabular form. It is difficult to get a sense of the numerical values from the figures/maps alone.

Experimental design

Overall, this is an interesting application of museum specimens to study spatio-temporal dynamics of animal movement. However, although the use of reference specimens to estimate an index of abudance may work in principle, I have some reservations about the methods used in this study. Specifically:

What methods were used to collect specimens (line 120)? How was it determined that all of the reference groups were collected using similar methods? Do the specific reference groups or collection methods influence AI? For example, mist-netting may favor certain groups (e.g. ground-dwelling passerines) compared to others (e.g. cuckoos). It would seem, therefore, that AI for PABU may increase or decrease depending on which groups were chosen as the reference. If true, than the three groups used in this study should be better justified and the sensitivity of results to this choice should be made explicit.

Collecting specimens surely comes with certain biases (as do all data) but these biases are not discussed. How sensitive are the AI estimates to sampling biases during collecting? For example, if collectors specifically target rare/unusual birds over common birds, could this inflate AI values (see comment on figure 2 below)?

The divisions used to partitian the annual cycle seem odd (lines 140-141). June is spring migration but April is winter? Aren't May and June breeding months? Isn't April (and possibly March) spring migration? If these are truly the periods of the MW PABU annual cycle, than more specific information (with citations) is needed to justify these divisions

How do the authors know that they were sampling only birds from the midwestern population of PABU? Figure 2 shows birds in the Yucatan which appear somewhat disjunct from the remaining birds. I've read that SE PABU may move through Cuba to winter in the Yucatan. Is it possible that these are from the southeastern population? If so, does the presence of SE PABU influence conclusions about the migration patterns of the MW population?

The tests listed (Lines 139-140) provide evidence that PABU are at higher abundances relative to reference specimens in certain regions during certain times of year. But another assumption of this study is that PABU move throughout the winter, i.e. will be at lower abundances in certain times of year. The conclusions of the paper would be strengthened by testing whether PABU were present in lower numbers than reference specimens during certain times of years (see comment below on AI values for April in figure 2 for one example of the need for additional tests).

One implicit assumption of the chi-square test approach is that the AI of the reference specimens remains stationary throughout the winter. Were the reference specimens also migratory species? Is it possible that the distribution of references species also changed over the course of the winter? If so, this could have a big impact of the conclusions about PABU. It would be good to test whether the AI of reference specimens changed over these periods.

I understand the rationale for sampling EVI values multiple times from the same location but this would appear to violate several assumptions of the t-test used below. In particular, the samples are clearly not independent of one another, which will lead to multicollinearity. The non-independence of the PABU sites also likely leads to different variances between the reference and the random samples, violating a central assumption of the t-test. It is difficult to tell whether these violations have an impact of the results of the current analysis (see comment on figure 3 below) but the authors should at least test whether their data meet the assumptions for the t-test, quantify whether these violations influence their results, and possibly use a more appropriate test.

Validity of the findings

Given the concerns about the experimental design listed above, it is difficult to judge the vailidy of the findings at this time. As presented, the findings showing semi-circular migration movements are not convincing, particularly without further tests showing low abundance in particular regions during particular times of year and possible accounting for the presence of southeastern PABU in eastern Mexico/Yucatan.

Problems with non-independence of the EVI values also prevent conclusive findings that PABU are tracking resource abundance. Until these tests are further justified or appropriate tests are used, it is difficult to judge whether the authors' conclusions are supported by the data.

Additional comments

General comments:

Although I agree that the use of NHCs is an interesting way to test population dynamics, the authors treat this techique as if it is the only alternative to directly tracking individuals. I would have like to seen a discussion of how this method complements or differs from other approaches, e.g. citizen science. In particular, the use of data from the eBird program is already providing highly detailed weekly distributions of many migratory bird species. The extensive spatial and temporal breadth of these data, and the resolution of the distrubution maps, would seem to exceed that of the NHC. What role do the authors see for their method when compared to citizen data?

Specific comments below and in the manuscipt:

Introduction:
Line 34: This seems like a bit of an overstatement. NHC may very well lack some of the biases of other data types but certainly come with their own important biases. Rather than making a big statement about lacking bias, I would recommend toning down this sentence a bit and linking to the specific biases biases that NHCs help overcome compared to other data types (which are listed in the next few sentences).

Line 47: This is not so obvious to me (or to other folks who do not think much about NHCs). Perhaps this sentence could be rephrased to simply state that NHCs have potential to provide insights in populations dynamics but only if collecting effort can be accounted for.

Methods:
Line 78: The distinction between the southeastern and midwestern breeding populations is important and central to this paper since the focus is on only the midwestern population. At some point in this paragraph, it should be made clear that the analysis and results is focused on midwestern PABU

Line 85: This is not a species-level question, correct? It is specific to the midwestern breeding population and should be stated as such

Line 99: I think the rationale behind AI needs to be more clear for readers that are not familiar with this method. It appears that the total number of specimens collected at a given place/time is taken as a measure of effort. Thus, if the focal species makes up a small percentage of this total, it is assumed to occur at low abundance. Is this correct? Either way, it would benefit readers to have a more detailed explanation of AI

Line 102: how is this expected number of specimens estimated? Is it just the sum of all specimens collected at a given place/time? That does not seem like an "expectation" to me since it already occurred.

Line 109: Again, please define what this means for readers not familiar with NHC. Does this mean that every specimen has a lat/long and date associated with it? Is it really 100% of specimens?

Line 129: How was density calculated? Is this different than AI?

Lines 148-149: This test comes out of the blue and it is not clear why the test for altitudinal effects is needed

Line 154: The link does not appear to be working.

Discussion:
Lines 231-232: This statement seems needlessly dismissive of tracking individual birds. There is no doubt that advances in tracking provide invaluable information about migration, habitat use, etc. for migratory birds. If anything, tracking is more powerful approach that the use of NHCs and will continue to be so as technology improves. This use of NHCs in this way is a cool advance but certainly complements rather than circumvents the use of direct tracking technologies.

Lines 243-244: Couldn't this be directly tested? see above

Line 253: If birds are tracking ephemeral resources (indicated by the association with EVI values), isn't it also possible that they would remain at relatively high densities, but just in different locations? This would seem to be a more common behavior for wintering grassland species

Line 288: This was not explicitly tested. Only the opposite was

Figures:

The text says April is winter. In figure 1 it is listed as both spring migration and winter. Which is it? And see above comment about the division of the annual cycle.

Figure 2:
The April map in figure 2 would seem to contradict the semi-cirucular migration route. It appears here that many (most?) birds are still in southern Mexico, rather than the gulf-Coast. This emphasizes the importance of testing for low abundance - abundance may be higher than reference specimens along the gulf coast, but also higher in other areas as well. Convincingly showing the proposed migration route requires showing both higher abundance along the gulf and lower abundance elsewhere.

What explains this (seemingly) high abundance point in NW Mexico in December? It seems like an outlier given the time of year and the November/January maps. If it is an outlier, this suggests that the AI approach may be sensitive to some aspect of the collection process (e.g. targetting unusual birds, changes in reference specimen abundance, etc)

Annotated reviews are not available for download in order to protect the identity of reviewers who chose to remain anonymous.

·

Basic reporting

The MS is well writen and to me is classicla scientific paper with nice structure.

Experimental design

In fact is not really experimental study, but using long term data from US museams. Data collecting are explained and interpreted in good way. I have one point on data collection patter and I put this directly on the MS.

Validity of the findings

Findings are extra-ordinary, mainly because is nice link between old fashioned museal data explained in light new remote sensing technologies. I think is anther argument to better exploration old, already collected data, than only to collect more and more new ones.

Additional comments

I put my comments and remarks directly on the MS.

---

## Round 0.2 · accepted · Accept

The reviewer and I feel that the revision has addressed all the main points raised in the first round of review. I believe that the manuscript is good to go, except for the small spelling changes suggested by the reviewer.

·

Basic reporting

It is the second turn, I see the current version is much better.

Experimental design

OK, but in fact it is a study based on museal collection

Validity of the findings

OK

Additional comments

I have two small comments. Is worth to add in the first sentence of introductiona also Hromada et al. 2003, and in introduction please change Hromanda to Hromada.